# *Candida dubliniensis* in Japanese Oral Microbiota: A Cross-Sectional Study of Six Geographic Regions in Japan

**DOI:** 10.3390/microorganisms12030525

**Published:** 2024-03-05

**Authors:** Tomoko Ohshima, Yoko Mukai, Hitoshi Watanabe, Keijiro Ohshima, Koichi Makimura, Takashi Komabayashi, Chul Ahn, Karen Meyer, Nobuko Maeda

**Affiliations:** 1Department of Oral Microbiology, School of Dental Medicine, Tsurumi University, 2-1-3 Tsurumi, Tsurumi-ku, Yokohama 230-8501, Japan; mukai-y@tsurumi-u.ac.jp (Y.M.);; 2Institute of Medical Mycology, Teikyo University, 359 Otsuka, Hachioji 192-0395, Japan; makimura@med.teikyo-u.ac.jp; 3Division of Comprehensive Oral Health—Endodontics, Adams School of Dentistry, University of North Carolina at Chapel Hill, Chapel Hill, NC 27599, USA; takashi.komabayashi@unc.edu; 4O’Donnell School of Public Health, University of Texas Southwestern Medical Center, Dallas, TX 75390, USA; chul.ahn@utsouthwestern.edu; 5Department of Dental Hygiene, Tsurumi Junior College, 2-1-3 Tsurumi, Tsurumi-ku, Yokohama 230-8501, Japan; meyer-k@tsurumi-u.ac.jp

**Keywords:** *Candida dubliniensis*, Japanese oral carriage rate, oral microbiota, ITS genotyping, antifungal drug susceptibility, secretory aspartic proteinase productivity, Japanese racial origin

## Abstract

Introduction: *Candida dubliniensis* was reclassified from the *C. albicans* genotype D, and reports show its frequent detection in HIV-positive individuals and easy acquisition of antifungal drug resistance. However, the oral carriage rate in healthy people and contribution to candidiasis in Japan is unclear. Methods: We conducted a cross-sectional survey of the *C. dubliniensis* carriage rate, performed genotyping and tested antifungal drug susceptibility and protease productivity. Specimens from 2432 Japanese subjects in six regions (1902 healthy individuals, 423 with candidiasis individuals, 107 HIV-positive individuals) were cultured using CHROMagar^TM^Candida, and the species was confirmed via 25S rDNA amplification and ITS sequences analyzed for genotyping. Results: The *C. dubliniensis* carriage rate in healthy Japanese was low in the central mainland (0–15%) but high in the most northerly and southerly areas (30–40%). The distribution of these frequencies did not differ depending on age or disease (HIV-infection, candidiasis). Genotype I, previously identified in other countries, was most frequent in Japan, but novel genotypes were also observed. Six antifungal drugs showed higher susceptibility against *C. albicans*, but protease productivity was low. Conclusions: Oral *C. dubliniensis* has low pathogenicity with distribution properties attributed to geography and not dependent on age or disease status.

## 1. Introduction

In Japan, where the aging of society is a considerable problem, pneumonia, including aspiration pneumonia, has been the third to fifth leading cause of death among people in recent years [1]. Genus Candida has been acknowledged as an important pathogen [2,3]. Moreover, oropharyngeal *Candida* infections are commonly diagnosed in human immunodeficiency virus (HIV)-infected individuals and AIDS patients. The most common cause of candidemia and oral candidiasis is *Candida albicans*, which grows not only as yeast cells but also as germ tubes and pseudohyphae. In 1995, an atypical *Candida* strain isolated from HIV-infected individuals showed a yeast-like, hyphal growth form, but a phylogenetical comparison of the nucleotide sequences of genome DNA with *C. albicans* showed that it belonged to a novel taxon. This reclassified strain was designated as *C. dubliniensis* [4]. Reports indicate that *C. dubliniensis* is recovered most often from the oral cavity of HIV-infected and AIDS patients [5], possibly because the azole-resistant strains were selected by researchers due to the introduction of fluconazole for the treatment of oral candidiasis in these patients during the HIV pandemic [6,7]. However, the oral carriage rate in healthy people is not known [8]. And it is not clear how *C. dubliniensis*, whose pathogenic factor has not been determined, contributes to general oral candidiasis compared to *C. albicans*. It is anticipated that secretory aspartic proteinase (SAP) production [9] and antifungal drug resistance [10,11] exert influence, and research into the mechanism and development of anti-SAP antibodies is underway. We believe that identifying the prevalence of *C. dubliniensis* in hosts would provide a clue regarding its role, since *Candida* must be present in the oral cavity for candidiasis to appear.

In the past three years, we have experienced a threatening situation in which the global COVID-19 pandemic has overwhelmed medical institutions. In an environment where the need for ICUs exceeded the actual number of ICUs in use, invasive fungal infections associated with COVID-19 have become a substantial complication affecting a significant number of critically ill patients hospitalized with COVID-19 [12,13]. Although *Candida* species were among the causative fungal pathogens, there are few reports of the involvement of *C. dubliniensis*. In Japan, there is only one reported case of fungemia acting as a rare endogenous infection in a severely ill patient with COVID-19 [14]; however, the pathogenicity of *C. dubliniensis* is unknown.

Ten years ago, a hospital in the Kyushu region of Japan analyzed the oral fungal flora of 52 patients with pseudomembranous oral candidiasis (disease group) and 30 healthy subjects (control group) by using an rRNA gene ITS (Internal Transcribed Spacer) and comprehensively performing regional sequence analysis [15]. The report stated that the prevalence of *C. albicans* was 55–65% in both groups, while the prevalence of *C. dubliniensis* was 0.5% in the control group and 12% in the disease group. In the disease group, the prevalence of *C. dubliniensis* decreased to 3% after treatment, so the authors viewed it as an important factor in the development of pseudomembranous oral candidiasis. However, the analysis’ results were based on a small number of subjects located in only one region of Japan. Therefore, it is necessary to expand the population area and number of subjects considered for analysis.

Japan is a long archipelago stretching in a northeast–southwest direction, with Honshu being the largest island and located in the center. The capital, Tokyo, is located almost at the center of the main island on the Pacific coast (Figure 1), and the other parts of the country are divided into 46 prefectures. It has been suggested that contemporary Japanese people are descendants of a group consisting of a mixture of the indigenous peoples (Ainu, Ryukyuan and mainlanders) who have been present in Japan since the Jomon era over 10,000 years ago, as well as immigrants who arrived between 2000 and several 100 years ago [16]. Recently, a study reported an analysis of SNPs (single-nucleotide polymorphisms) in each of Japan’s 47 prefectures (including Tokyo) [17]. When a principal component analysis was performed, the first principal component reflected the genetic distance between Okinawa prefecture and most other prefectures. However, the Kyushu region and the Tohoku region (including Aomori prefecture) were genetically close to Okinawa prefecture. The second principal component was significantly correlated with the latitude and longitude of the prefectures (latitude: *p*-value = 3.21 × 10^−12^; longitude: *p*-value = 2.38 × 10^−14^). This report suggests that differences in the degree of interbreeding between Jomon people in each prefecture and immigrants from the continent, as well as geographical location, are the main factors that have created regional genetic differences among mainlanders. Therefore, we anticipated the possibility that the colonization rate and isolation rate of resident fungi might be related to the diversity of the Japanese host population. To conduct an investigation, we selected six regions (Figure 1), where location would reflect the diversity of the host population.

The purpose of this study was to collect information about the prevalence and genotype of *C. dubliniensis* in healthy Japanese people from multiple regions and their tendency toward antifungal drug resistance and proteinase producibility to compare these data with those of HIV-infected patients and oral candidiasis patients. We conducted an epidemiological survey to detect the oral carriage rate in more than 2000 people, ranging from juveniles to the elderly, in six regions of Japan and performed genotyping of *C. dubliniensis* isolates.

## 2. Materials and Methods

### 2.1. Subjects

The Ethics Committee of the Tsurumi University School of Dental Medicine approved the protocol for this study, and all subjects gave informed consent prior to inclusion in this study (Approval numbers: 031-114-117-118-133-327). Three groups (the healthy group, candidiasis group and HIV group) were investigated, with a minimum sample size of 100 subjects in each group, so that the maximum standard error for the prevalence rate was less than 5%. We included 1902 healthy subjects (healthy group) from six geographic regions of Japan (Table 1, Figure 1), who were recruited from April 2001 to July 2010, with the exclusion criterion of being hospitalized, institutionalized or requiring special care for daily activities. Aomori faces the Sea of Japan and the Pacific Ocean at the northernmost point of the mainland. Niigata is located in the central mainland, facing the Sea of Japan. Nagano is also in the central mainland, surrounded by mountains. Tokyo/Kanagawa is centrally faces the Pacific Ocean. Yamaguchi faces the Inland Sea at the southern end of the mainland. The island of Okinawa is found off the southern coast in the East China Sea (Figure 1).

The candidiasis group consisted of 423 patient subjects who visited the Dry Mouth Clinic at Tsurumi University Dental Hospital from January 2003 to May 2007, with the inclusion criteria being a candidiasis diagnosis and the detection of candida via a mucosal swab test. The HIV-positive subject group (HIV group) was composed of 107 individuals referred to the Department of Oral and Maxillofacial Surgery, Tokyo Medical University Hospital, from September 2001 to May 2004, with the inclusion criterion being a diagnosis with the HIV RNA copy load determined via flow-cytometry and reverse transcribed (RT)-PCR. The exclusion criterion for the subjects was treatment with antifungal medication within one month.

### 2.2. Collection of Candida Specimens and Culturing

The specimens were collected and cultured according to previously reported methods [18]. In brief, each sample was obtained from the subject’s tongue by a dentist, who firmly wiped the dorsal portion exactly 10 times with a sterile cotton-wool swab (Code 06525, Nissui, Tokyo, Japan), which was immediately inoculated onto a CHROMagar^TM^Candida (ChromAgar Co., Ltd., Paris, France), and colonies formed after 48–72 h, which were used for pure culturing on a Sabouraud dextrose medium. DNA was extracted from the fungal cells, and the remaining cells were collected and stored at −80 °C until susceptibility testing was performed. The DNA samples were stored at −20 °C and analyzed within 1–2 months to identify the bacterial species.

### 2.3. Species Determination

Two days after their cultivation at 30 °C, the isolates of *Candida* spp. were identified according to their colony morphology and color on plates or the api20C AUX^®^ system (bioMerieux, Marcy l’Etoile, France).

### 2.4. Identification of C. dubliniensis and Genotyping of C. albicans

The *C. albicans* genotypes and *C. dubliniensis* were determined via the PCR amplification of the 25S rRNA gene (rDNA) region [19]. As only *C. albicans* and *C. dubliniensis* appeared as green colonies on CHROMagarTMCandida, one green colony per agar plate was picked up, and the genomic DNA was extracted using a PrepManTMUltra (Applied Biosystems, Foster City, CA, USA). The PCR was performed as described previously [19], with a slight modification enabled by using the HotStarTaq^®^ Master Mix (Qiagen GmbH, Hilden, Germany) and the primer pairs CA-INT-L: 5′-ATAAGGGAAGTCGGCAAAATAGATCCGTAA-3′ and CA-INT-R: 5′-CCTTGGCTGTGGTTTCGCTAGATAGTAGAT-3′ under the following conditions: the first denaturation was performed at 95 °C for 15 min, followed by 30 cycles at 94 °C for 1 min, 65 °C for 1 min and 72 °C for 2.5 min; the final extension was performed at 72 °C for 8 min.

The DNA sequence analysis from the ITS1 to the ITS2 region of *C. dubliniensis* was performed using the ABI Prism BigDye Terminator V3.1 Cycle Sequencing Kit (Applied Biosystems). The base sequence of each reaction product was read via an automatic sequencer 3730X DNA analyzer (Applied Biosystems). The sequence homology was confirmed using the BLAST search system.

The ITS sequences were aligned with a Clustal W 1.6.6 computer program [20]. A cluster analysis was then performed using the neighbor-joining (NJ) method with the NJPLOT program [21]. Bootstrap [22] analysis with Clustal W was performed for 1000 random samples from the multiple alignment to provide a measure for the robustness of the tree, given the data set and methods used. The tree was rooted by using *Candida albicans*, DDBJ/EMBL/GenBank accession no. AY939786, as an outgroup.

### 2.5. Antifungal Drug Susceptibility Test

The Yeast-Like Fungus FP ‘Eiken’ (Eiken Chemical Co., Ltd., Tokyo, Japan), developed in accordance with the M27-A3, 2008 Code of Practice, was used to test six antifungal drugs (AMPH-B, Amphotericin-B; 5-FC, Flucytosine; FLCZ, Fluconazole; ITZ, Itraconazole; MCZ, Miconazole; MCFG, Micafungin) for the antifungal susceptibility of Candida isolates. In summary, the tested Candida was cultured overnight in Sabouraud dextrose medium (Eiken Chemical Co., Ltd.) and suspended in saline (the fungal solution concentration was adjusted to the same turbidity as the McFarland standard turbidity solution 0.5, and the fungal solution for inoculation was diluted 10 times with sterile saline). The fungal solution (approx. 0.0005 mL of 1–2 × 10^5^ colony-forming unit/mL) was inoculated in each well with the Yeast-Like Fungus FP ‘Eiken’, except for the negative control (NG-C), and cultured at 35 °C for 24 h and 48 h. The turbidity was confirmed visually, and the minimum inhibitory concentration (MIC) was determined.

Amphotericin B completely inhibits growth (at the same level as NG-C). Other drugs are based on 50% growth inhibition (IC_50_) compared to the positive control (PG-C), and growth inhibition below that level was determined. IC_50_ wells were prepared by placing 0.05 mL of the culture solution of the inoculated fungal cells into an empty well and adding 0.05 mL of sterile purified water or sterile saline. A judgment was made by visually comparing the turbidity of each well with that of the IC_50_ well.

### 2.6. Secretory Aspartic Proteinase (SAP) Productivity Test

The method of Lam, M. et al. [23] was slightly modified. In summary, the tested Candida were cultured overnight in Sabouraud dextrose medium (Eiken Chemical Co., Ltd.), and then 1/2 inoculation loop grown shaking was performed in 10 mL of a Yeast Carbon Base (YCB) medium (BD Difco™, Becton, Dickinson and Company, Franklin Lakes, NJ, USA) supplemented with 0.2% BSA (FUJIFILM Wako Pure Chemical Corporation, Osaka, Japan) at 25 °C for 3 days. After centrifugation, 100 µL of the culture supernatant was mixed with 100 µL of a 1% BSA solution dissolved in 0.1 M Na-Citrate buffer (pH 3.2) as an enzyme substrate and incubated at 37 °C for 6 h. The reaction was stopped by adding 1 mL of 10% TCA (trichloroacetic acid), the solution was kept on ice for 1 h before being centrifuged, and the supernatant’s absorbance at 280 nm was measured using a spectrometer (Ultrospec4000, GE Healthcare/Amersham, Chicago, IL, USA).

### 2.7. Statistical Analysis

The statistical analysis was performed using SPSS 14.0J (SPSS Japan, Tokyo). A chi-square test with the Yates adjustment or Fisher exact tests were conducted to investigate whether the prevalence rates (%) of the candida species and *Candida* A, B, C and D genotypes were significantly different between regions and between healthy subjects and patients. The SAP enzymatic level among multiple groups was analyzed via the Kruskal–Wallis test and between two groups via the Mann–Whitney U test. Statistical significance was set at *p* < 0.05.

## 3. Results

### 3.1. Prevalence of Oral Candida and Candida Species in Healthy Subjects in Different Geographic Regions and with Different Health Conditions

The number of subjects in each group and carriage rate of the *Candida* species are summarized in Table 1. A total of 2432 Japanese, consisting of 1902 healthy subjects of varied age from six regions in Japan (Figure 1) and 530 patient subjects in the Tokyo/Kanagawa region, were investigated in this study. As shown in Table 1, the *Candida* prevalence rate in the oral cavity tended to increase with age. The juveniles had a lower prevalence of *Candida* than the adults in most regions, except for Okinawa.

The albicans group (*C. albicans* and *C. dubliniensis*) was the most common species in all the groups, followed by *C. glabrata* (Table 1). A low prevalence of varieties other than the three *Candida* species, *C. albicans*, *C. dubliniensis* and *C. glabrata*, was observed in the healthy groups of all regions, although there was no significant correlation with geographic region for these other varieties.

### 3.2. Proportion of Each Genotype of C. albicans and C. dubliniensis

The genotypes of *C. albicans* were classified into three types (A, B and C) depending on the differences in the group I intron sequence of the 25S rRNA gene (rDNA); *C. dubliniensis* was identical to genotype D. To confirm the species determination of *C. dubliniensis*, a sequence analysis of the DNA region from ITS1 to ITS2 was carried out and checked using the BLAST search systems. The most common genotype of *C. albicans* in both the healthy adults from all geographic regions and the patient subjects was type A. The least common genotype in most groups of adults was type D (*C. dubliniensis*) (Figure 2). However, in Aomori and the remote island region of Okinawa, *C. dubliniensis* (type D) was the second-most common type at 30~40% in the albicans group, and the distribution patterns of the four genotypes in these two regions were significantly different from those in the other regions (Figure 2). As for the patient groups in the Tokyo/Kanagawa region, the genotype distribution patterns were not different from those of the healthy adults in the same region, but they were different from those in other regions, particularly Aomori and Okinawa (Figure 2, *p* < 0.05). The differences in the four-genotype distribution were statistically analyzed using the chi-square or Fisher exact test. The *p*-values of the significant differences between each of the two groups are shown as follows: for healthy adults in Aomori, the *p*-value was *p* = 0.046 for Niigata, *p* < 0.001 for Nagano and *p* = 0.031 for Tokyo/Kanagawa (*p* = 0.059 for Yamaguchi). For Okinawa, the value was *p* < 0.001 for Niigata, *p* < 0.001 for Nagano, *p* = 0.040 for Tokyo/Kanagawa and *p* = 0.005 for Yamaguchi. For the patient group and healthy subjects, as well as for HIV or the candidiasis patients, the *p*-value was *p* = 0.040 or *p* < 0.001 for Aomori and *p* < 0.001 for Okinawa, but there were no significant differences for Niigata, Nagano, Tokyo/Kanagawa and Yamaguchi (Figure 2).

In the juveniles, there was an overwhelming prevalence of genotype A but no detection of type D (*C. dubliniensis*) in the Tokyo/Kanagawa region. However, in Okinawa and Aomori, *C. dubliniensis* was the most common or the second-most common genotype (Figure 3). The prevalence of *C. dubliniensis* in Okinawa and Aomori was apparently higher than that in Tokyo/Kanagawa. Differences in the four-genotype distribution between the two regions in each age group were statistically analyzed via the chi-square or Fisher exact test. The distribution patterns of the four genotypes in juveniles were significantly different in these two regions compared to those in the Tokyo/Kanagawa region (***: *p* < 0.001).

To classify *C. dubliniensis* isolates via genotyping on the DNA sequences from ITS1 to ITS2 of the intervening 5.8S rRNA gene15, 122 strains of *C. dubliniensis* (93 from healthy subjects and 29 from patient subjects) isolated in this survey (Table 2) were analyzed via DNA sequencing, multiple alignment and the drawing of the phylogenetical tree via the neighbor joining method using Clustal W software. The results showed that the Japanese *C. dubliniensis* strains were divided into nine phylogenic groups (genotypes) (Figure 4). Most strains, both from healthy and patient subjects, were classified into genotype I, which was identical to genotype 1, as previously reported [24]. However, only three strains were classified into genotype 2, and none were classified into genotypes 3 or 4 (Table 2). On the other hand, novel genotypes (II to V and IX) were found in the Japanese strains. Genotype II was near to genotype I, genotype III was near to genotype IV, genotype V was near to genotype VI (identical to genotype 3 in the previous report [15]), and genotype IX was near to genotype VIII (identical to genotype 2 in the previous report [16]) (Figure 4).

### 3.3. Antifungal Drug Susceptibility

For all five antifungal agents examined, the MIC range and the values of the MIC_50_ and MIC_90_ inhibitors of strains derived from HIV-negative and HIV-positives individuals were lower for *C. dubliniensis* than for the three genotypes (A), (B) and (C) of *C. albicans*. Comparing the origins of the fungal strains between HIV-negative and HIV-positive individuals revealed almost no difference (Table 3).

### 3.4. Secretory Aspartic Proteinase (SAP) Productivity

SAP production in *C. dubliniensis* was significantly lower than that in the three *C. albicans* genotypes (A), (B) and (C) (Figure 5). Furthermore, when only comparing strains derived from HIV-negative patients, *C. dubliniensis* showed a slightly but significantly lower value than *C. albicans* (*p* < 0.001).

## 4. Discussion

*C. dubliniensis* reclassified from *C. albicans* has recently gained attention because of its association with candidiasis in HIV-positive individuals and its resistance to antifungal agents [6]. These similar species are best identified based on specific differences in their genomic DNA [19,25]. The purpose of this study was not only to identify *Candida* species and genotyping via DNA sequence analysis but also to compare antifungal drug susceptibility and protease producibility, which are traits related to pathogenicity. Therefore, we decided to conduct an investigation using a culture method to obtain viable isolates.

The proportions of the three genotypes of *C. albicans* and *C. dubliniensis* have previously been reported in hundreds of isolates obtained from geographically diverse regions [19]. In these previous reports, genotype A was the most common; B and C were found at 10 to 30%, while *C. dubliniensis* was found at a low percentage (0.5–2%) among these *C. albicans* family (*C. albicans* and *C. dubliniensis*) isolates. The carriage rate of *C. dubliniensis* in the oral cavity of various hosts reported by country was 3.98% in HIV-positive Venezuelans [26], 3.5% in HIV-negative healthy Irish people [8], 2.25% in isolates from hospitalized Italians with various diseases [27] and 0–16.4% among HIV-negative and HIV-positive black and white South Africans [28]. Thus, the global prevalence of *C. dubliniensis* appeared to be quite low. In Japan, only one systematic study has previously been conducted for the genetic subtyping of *C. albicans* and *C. dubliniensis* in 301 clinical isolates collected from three Japanese regions [25]. The researchers identified only five isolates as *C. dubliniensis*, and those isolates were only derived from diseased hosts in a limited area of Japan; they did not determine the carriage ratio in Japanese individuals [25].

In the present study, we selected six regions (Table 1, Figure 1) with different geographical features. To generate survey participants, we used one green colony in CHROMagar*Candida*^TM^ from each of the 1155 subjects selected from among 1902 healthy subjects and 530 patient subjects. We identified 981 isolates of *C. albicans* and 174 isolates of *C. dubliniensis*. In our investigation, if both *C. albicans* and *C. dubliniensis* were present on the ChromAgar^TM^Candida medium of one subject, the one with a higher proportion was preferentially determined. Therefore, although it is possible that the subject was judged to be negative even though it was positive, the reverse should not have happened, so it is impossible that the detected frequency was higher than the actual one. Then, the result of the importance of focusing on areas with high detection frequencies seems not to be affected. Also, since the used method is different from those of previous research reports, direct comparison is questionable, but there seems to be no problem with regional comparisons within this study. But, the coexistence ratio of the two fungal species was unknown, and we regarded this as a limitation of this study.

The prevalence of *C. dubliniensis* appeared to vary by region, being more prevalent in Aomori and Okinawa than in the middle of the mainland among HIV-negative healthy subjects (Figure 1, Figure 2 and Figure 3), and it was explained as the “*C. dubliniensis* carriage rate in healthy Japanese was low in the central mainland (0–15%), but high 26 in the most northern and southern areas (30–40%)”, suggesting that geography affects the distribution of *C. dubliniensis*. Xu et al. noted geographical differences in the prevalence of human *Candida* species and diversity in oral bacterial floras, citing examples from eastern North America and China [29]. In another report, the prevalence of *C. dubliniensis* correlated more with race in South Africa than with HIV infection status or geographical location [28]. In Japanese history, at least three distinct ethnic groups, the Ainu, the Ryukyuan and the Wajin, contributed to the formation of the present Japanese population [30]. The Ainu and the Ryukyuan are considered to be aboriginal native populations that have principally lived in the north (Hokkaido and Tohoku) and the south (Okinawa), respectively, since the prehistoric Jomon period more than 10,000 years ago [31]. The Wajin are considered to be descendants of postneolithic Yayoi period migrants (300 BC-600 AD) from China and South Korea to the mainland, presently forming the largest part of the Japanese population [32]. This population history of the Japanese is called the “dual structure model” [30,33], which is supported not only by genetic and morphological studies [30,31,32,34] but also by HTLV-1 infection studies [35,36]. There is a high frequency of HTLV-1 infection in the southern area of Japan, but the second-highest frequency occurs in the northern area (Hokkaido and Tohoku area). The results of this study of the high detection rate of *C. dubliniensis* in the southern islanders and most people in Honshu (Figure 1, Figure 2 and Figure 3) also seems to agree with this phenomenon and theory.

Although the frequency and number of *Candida* species generally increased with age in healthy people [37] or with immunosuppressive conditions in diseased people, it is not known whether the proportion of each *Candida* genotype changes with age or health condition. Based on the data in this study, it is possible that the proportion of the genotype does not change with age or even with HIV infection or candidiasis but with geographic region (Figure 1, Figure 2 and Figure 3). Some pathogenic microorganisms are known to have genetic and geographic variations in Japan. In particular, each genotype of HTLV-1 [35,36], human simplex virus [38] and human JC polyoma virus [39] isolates from the Ainu in Hokkaido was clustered in the same phylogenic tree as the Okinawan isolates. We also assessed the genotype distribution of *C. dubliniensis* isolated from the six regions to identify any micro-evolutional changes, as reported by Gee et al. [24], but we found that most strains were of the same genotype (genotype I) (Table 2), suggesting that a single, original strain of *C. dubliniensis* (genotype I) may have spread during an early era in Japanese history.

Since *C. dubliniensis* was first isolated and identified from HIV-infected individuals, it was assumed to be frequent in HIV-infected individuals [5] and highly pathogenic with a tendency toward antifungal drug resistance and protease (SAP) production [6,7]. However, recent reports have shown that *C. dubliniensis* is less pathogenic than *C. albicans* [40,41,42]. In this study, we conducted antifungal susceptibility tests and SAP productivity measurements for confirmation of the Japanese isolates of *C. dubliniensis*.

When we compared the susceptibilities mainly based on the MIC breakpoints shown via CLSI and EUCAST [43,44], multiple resistant strains were found in *C. albicans* but not in *C. dubliniensis* strains derived from both HIV-positive patients and non-patients (Table 3). Regarding SAP productivity, *C. dubliniensis* showed significantly lower values than *C. albicans* (Figure 5). Therefore, the pathogenicity of the Japanese *C. dubliniensis* strains is considered to be weak.

Since *C. dubliniensis* has significantly less virulence compared to *C. albicans* [7], it seems to co-exist with its hosts for a long period if colonized as a latent member of oral microbiota. Moreover, we found that minor novel genotypes (II, III, IV, V and IX) of *C. dubliniensis* spread micro-evolutionally in Japan (Table 2). Genotypes III and IV are close clades (Figure 4), and among healthy people, these genotypes were found only in isolates from Okinawa (Table 2). As genotypes VI or VII (reported by Gee et al. [24] as genotypes 3 or 4) were not found in our specimens, they are considered to be non-Japanese-type clades.

## 5. Conclusions

The *C. dubliniensis* carriage rate in healthy Japanese was low in the central area of the mainland (0–15%) but high in the most northerly and southerly areas (30–40%). The distribution of these frequencies did not differ depending on age or disease (HIV infection, candidiasis) but can be attributed to geography.

The clinical implication of this study was the observation that oral *C. dubliniensis* has low pathogenicity concerning antifungal drug resistance and protease producibility. The significant low virulence might be an indication of long-lasting residence in the human oral microbiota.

In order to elucidate the host specificity of *C. dubliniensis,* further analysis of detailed host factors is necessary, including environmental and genetic variations affecting *C. dubliniensis* colonization.

## Figures and Tables

**Figure 1 microorganisms-12-00525-f001:**
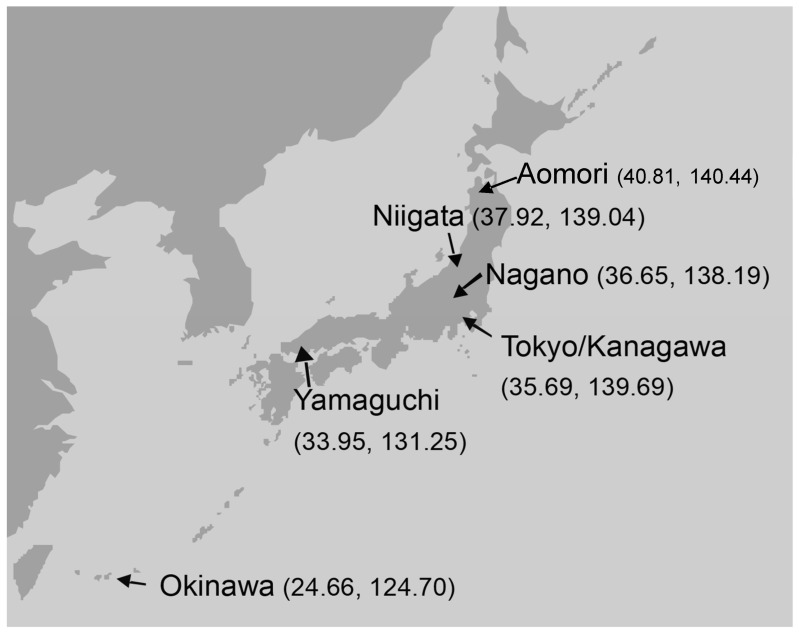
Six Japanese regions surveyed in this study. The mapping position of each location is shown in parentheses (north latitude, east longitude).

**Figure 2 microorganisms-12-00525-f002:**
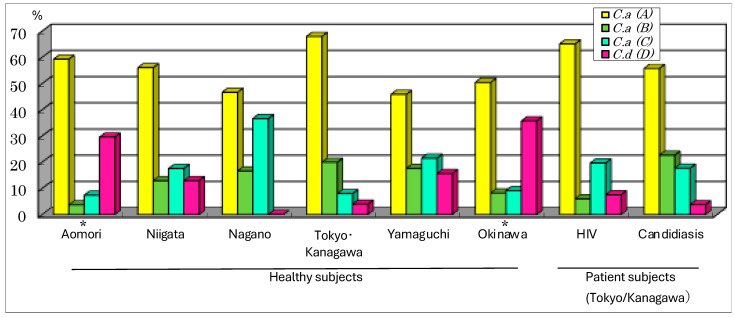
Genotypes of *Candida albicans* and *C. dubliniensis* in adults in six regions. The carriage rates of each genotype of *C. albicans* (genotypes A, B and C) and *C. dubliniensis* (genotype D) in healthy adults of six regions and in patient adults are displayed. The age and number of each subject group are indicated in Table 1 (the chi-square or Fisher exact test for Tokyo/Kanagawa, *: *p* < 0.05).

**Figure 3 microorganisms-12-00525-f003:**
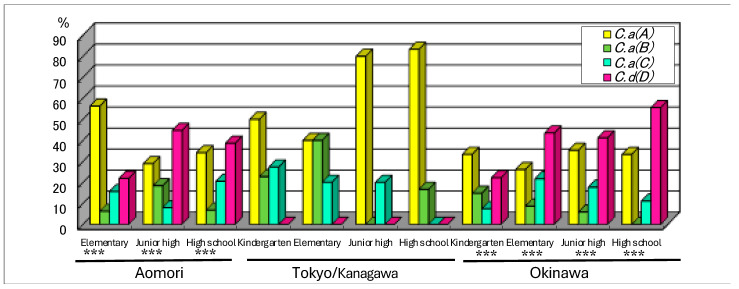
Genotypes of *C. albicans* and *C. dubliniensis* in juveniles from the Tokyo/Kanagawa, Okinawa island and Aomori regions. The carriage rates of each genotype of *C. albicans* and *C. dubliniensis* in juveniles from the regions of Tokyo/Kanagawa and Okinawa island or Aomori were compared (differences of the four-genotype distribution was compared with the chi-square or Fisher exact test, ***: *p* < 0.001).

**Figure 4 microorganisms-12-00525-f004:**
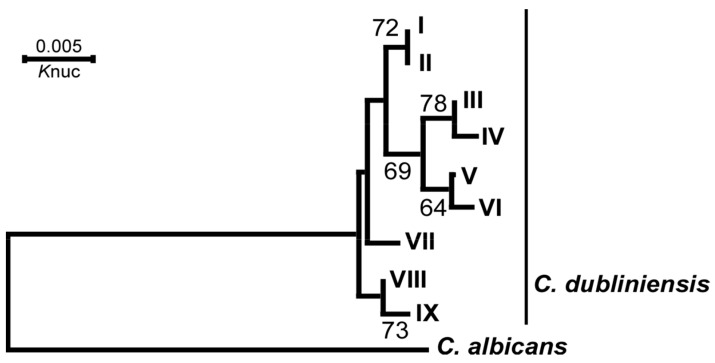
Phylogenetical plot of *C. dubliniensis* isolated from healthy individuals of various ages in six regions and the patient group of Tokyo/Kanagawa using a cluster analysis based on ITS DNA sequences. The ITS sequences were classified into nine genotypes (genotypes I to IX), and genotypes 1 to 4 were derived from a previous report [24]. The tree was drawn using the NJ method and rooted with *C. albicans* as an outgroup. The bootstrap samplings, derived from 1000 samples supporting the interior branches, are indicated. The numerical value in the branch shows the genetic distance from *C. albicans*.

**Figure 5 microorganisms-12-00525-f005:**
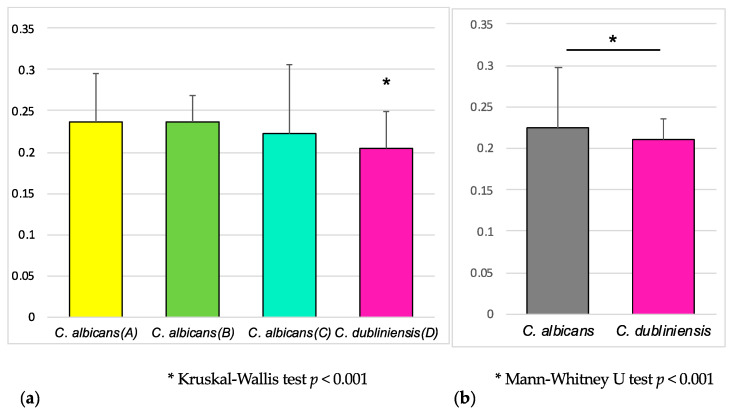
SAP activity of the isolated *C. albicans* and *C. dubliniensis*: (**a**) SAP activity of the isolated *C. albicans* of each a–c genotype (*n* = 42, 41 and 43, respectively) and *C. dubliniensis* (*n* = 43); (**b**) SAP activity of the *C. albicans* (*n* = 26) and *C. dubliniensis* (*n* = 11) only isolated from HIV-positive patients.

**Table 1 microorganisms-12-00525-t001:** The prevalence of *Candida* species in healthy subjects and patients in Japan.

Groups	Total Number of Individuals	Average Age (Range)	*Candida* Positive Rate %						
			Total *Candida*	*Albicans* Group (*C. a* and *C. d*)	*C. g*	*C. k*	*C. p*	*C. t*	*Candida* spp. (Others)
**Healthy subjects**									
Aomori									
Elementary school	133	7.2 (6–8)	28.6	24.1	3.8	0.8	0	0	0.8
Junior high school	135	12.1 (12–15)	29.6	26.7	3.7	0	0	0	0
High school	136	15.1 (15–17)	39.0	36.8	2.2	0	0	0.7	0
Adult	60	55.5 (20–85)	56.7	45.0	31.7	6.7	6.7	3.3	0
Sub-total	464		35.6	30.4	6.9	1.1	0.9	0.7	0.2
Niigata									
Adult	490	60.7 (20–75)	56.9	48.6	20.2	1.6	4.5	2.9	3.7
Nagano									
Adult	108	49.3 (24–80)	37.0	27.8	6.5	0.9	5.6	1.9	1.9
Tokyo/Kanagawa									
Kindergarten	121	4.2 (3–5)	28.9	17.4	10.7	0	0	0	3.3
Elementary school	38	9.2 (10–12)	13.2	13.2	0	0	0	0	0
Junior high school	38	13.4 (13–14)	26.3	26.3	0	0	2.6	0	0
High school	32	15.8 (15–17)	25.0	25.0	0	3.1	0	0	0
Adult	130	32.8 (20–76)	31.5	20.2	6.9	1.6	0	0	0
Sub-total	359		27.6	19.5	6.1	0.8	0.3	0	2.2
Yamaguchi									
Adult	129	48.4 (23–81)	45.0	41.1	9.3	3.1	3.1	0.8	2.3
Okinawa(isolated island)									
Kindergarten	56	3.0 (1–5)	55.4	50.0	10.7	0	0	0	0
Elementary school	55	9.5 (6–12)	58.2	45.5	10.9	0	1.8	0	0
Junior high school	51	13.3 (13–15)	39.2	35.3	3.9	3.9	3.9	0	0
High school	13	17.6(16–18)	69.2	53.8	15.4	0	0	7.7	0
Adult	177	47.7 (21–74)	58.2	50.3	14.1	5.6	5.6	0.6	1.7
Sub-total	352		55.4	47.4	11.7	3.4	3.7	0.6	0.9
**Patient subjects** (Tokyo/Kanagawa)									
HIV positives * (adult)	107	37.2 (21–71)	63.6	61.7	5.6	0.9	1.9	0.9	1.9
Oral candidiasis (adult)	423	66.8 (21–97)	100	92.2	46.6	0.8	13.7	7.1	9.9

The adults are 20 years old or older. *C. a*, *C. albicans*; *C. d*, *C. dubliniensis*; *C. g*, *C. glabrata*; *C. k*, *C. krusei*; *C. p*, *C. parapsilosis*; *C. t*, *C. tropicalis*. *: At the time of sample collection in 2004, the number of newly reported HIV infections in Japan was 780 (698 men, 82 women), the cumulative number of HIV infections per 100,000 people was 5140, the number of AIDS patients was 385 (344 men, 41 women), and the number of patients per 100,000 people was 2568 (https://idsc.niid.go.jp/iasr/26/303/tpc303-j.html#: accessed on 7 February 2024).

**Table 2 microorganisms-12-00525-t002:** Genotypes of *C. dubliniensis* isolates based on ITS ribosomal DNA sequences.

Genotype		Number of Strains		
In This Study	by Gee et al., 2002 [24]	From Healthy Volunteers	From Patient Subjects	Sub-Total
I	1	80 ^(a)^	23 ^(e)^	103
II *	N/I	1 ^(b)^	0	1
III *	N/I	6 ^(c)^	1 ^(f)^	7
IV *	N/I	1 ^(c)^	0	1
V *	N/I	3 ^(d)^	3 ^(f)^	6
VI	3	0	0	0
VII	4	0	0	0
VIII	2	1 ^(b)^	2 ^(f)^	3
IX *	N/I	1 ^(b)^	0	1
Total		93	29	122

*: new finding in this study; N/I: not identified; ^(a)^: details were as follows—11 strains from Aomori, 23 strains from Niigata, seven strains from Nagano, one strain from Tokyo/Kanagawa, eight strains from Yamaguchi and 30 strains from Okinawa; ^(b)^: derived from Niigata; ^(c)^: derived from Okinawa, ^(d)^: two from Niigata and one from Aomori, ^(e)^: 18 strains derived from candidiasis patients and five from HIV positive patients, ^(f)^: derived from candidiasis patients.

**Table 3 microorganisms-12-00525-t003:** Antifungal drug susceptibility of *C. albicans* and *C. dubliniensis* isolated from HIV-negative and HIV-positive individuals.

		Amphotericin-B (µg/mL)			Flucytosine (µg/mL)			Fluconazole (µg/mL)			Itraconazole (µg/mL)			Miconazole (µg/mL)			Micafungin (µg/mL)		
	n	MIC Range	MIC_50_	MIC_90_	MIC Range	MIC_50_	MIC_90_	MIC Range	MIC_50_	MIC_90_	MIC Range	MIC_50_	MIC_90_	MIC Range	MIC_50_	MIC_90_	MIC Range	MIC_50_	MIC_90_
From HIV negatives																			
*C. albicans* (A)	30	0.25–1.0	0.5	1.0	0.125–0.5	0.125	0.125	0.25–128	4.0	128	0.015–16	16	16	0.06–64	4.0	64	0.03 ^(2)^	0.03	0.03
*C. albicans* (B)	12	0.13–1.0	0.5	1.0	0.125	0.125	0.125	0.125–128	0.5	4.0	0.03–1.0	0.06	0.5	0.06–1.0	0.125	0.25	0.03 ^(3)^	0.03	0.03
*C. albicans* (C)	12	0.125–1.0	0.25	1.0	0.125	0.125	0.125	0.25–128	0.5	128	0.03–128	0.06	1.0	0.06–4.0	0.125	2.0	0.03	0.03	0.03
*C. dubliniensis* (D)	20	0.03–0.5	0.25	0.5	0.125 ^(1)^	0.125	0.125	0.125–0.5	0.25	0.5	0.015–0.125	0.06	0.125	0.06–0.125	0.06	0.125	0.03	0.03	0.03
From HIV positives																			
*C. albicans* (A)	38	0.125–1.0	0.5	1.0	0.125–128	0.125	0.5	0.125–128	32.0	128	0.03–16	8.0	16	0.06–64	8.0	64	0.03–2	0.03	0.03
*C. albicans* (B)	7	0.25–0.5	0.5	0.5	0.125	0.125	0.125	0.125–128	0.25	128	0.03–16	0.06	8.0	0.06–16	0.06	8.0	0.03	0.03	0.03
*C. albicans* (C)	13	0.5–1.0	0.5	1.0	0.125–0.25	0.125	0.125	0.125–128	1.0	128	0.03–16	0.125	16	0.06–64	0.25	64	0.03	0.03	0.03
*C. dubliniensis* (D)	5	0.06–0.5	0.5	0.5	0.125	0.125	0.125	0.25–1.0	0.5	1.0	0.06–0.125	0.06	0.125	0.125	0.125	0.125	0.03	0.03	0.03

^(1)^ *n* = 19 ^(2)^ *n* = 29 ^(3)^ *n* = 11. MIC of each genotype of *C. albicans* (genotypes A, B and C) and *C. dubliniensis* (genotype D) from HIV negatives and positives was shown.

## Data Availability

The nucleotide sequence data of the novel genotypes of *C. dubliniensis* reported in this study are available in the DDBJ/EMBL/GenBank databases under the accession numbers AB292451, AB292452, AB292453, AB292454, AB292455, AB292456, AB292457, AB292458, AB292459, AB292460 and AB292461.

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
