# Peer review of "Candida dubliniensis in Japanese Oral Microbiota: A Cross-Sectional Study of Six Geographic Regions in Japan"

_microorganisms, 2024, doi:10.3390/microorganisms12030525_

Round 1
Reviewer 1 Report
Comments and Suggestions for Authors
This is an interesting and relevant study about the prevalence of a neglected pathogen in Japan, I congratulate the authors for placing the spot on an underestimated Candida species. I have the following comments though for improvement:
Please explain whether strains were processed at the collection time or they were kept in refrigeration/freezer and analyzed afterward. Some samples were collected more than twenty years old! In two decades, the way these strains are selected in Chromagar and the molecular methodology for genotypification have significantly evolved. If they are part of a collection that was analyzed later, this could be a bias, particularly in the antifungal susceptibility test.
Since only one green colony was taken for analysis, some samples may have the presence of both Candida albicans and Candida dubliniensis. Since co-infection was not sought as part of the study, this has to be acknowledged as a study limitation.
The abstract contains the following sentence: "C. dubliniensis carriage rate in healthy Japanese was low in the central mainland (0–15%), but high 26 in the most northern and southern areas (30–40%)."
This is a relevant observation that deserves elaboration in the discussion section.
Finally, the abstract is wordy and should be improved.
Comments on the Quality of English LanguageThere are some typos and names that should be in italics but this can be fixed during the galley proofs.
Author Response
Thank you very much for taking the time to review this manuscript. Please find the detailed responses for your valuable comments, in an attached file.

Reviewer 2 Report
Comments and Suggestions for Authors
Dear Authors,
the study is interesting but some additional aspects could improve its quality.
Could you please provide some epidemiological data on HIV-infected patients in Japan during this period.
How was performed the sample calculation?
Why gender of population was not included in this study?
Could you please provide short ethnic characterisation of Japanee pupulation in choosen region in the M&M section.
Lack of clinical implications of your study.
Lack of conclusions section.
Please add necessary information to the manuscript.
Author Response

(The authors gave the same response as above.)

Reviewer 3 Report
Comments and Suggestions for Authors
1. The introduction to the article should be completely redone. Firstly, the links to the text are not correct.
For example, at the beginning of the introduction there is a phrase: “During the last two to three decades, the number of opportunistic fungal infections among immunocompromised hosts, particularly hospitalized patients, and the rate of bloodstream infections (BSIs) occurring as nosocomial transmissions of fungal pathogens and disseminated candidiasis has increased by almost five to ten times worldwide [1-2], a situation that threatens human societies [2-3].”
References: 1. Pfaller, M.; Wenzel, R. Impact of the changing epidemiology of fungal infections in the 1990s. Eur J Clin Microbiol Infect Dis. 1992, 11, 287-91.
2. Fisher-Hoch, S.P.; Hutwagner, L. Opportunistic candidiasis: an epidemic of the 1980s. Clin Infect Dis. 1995, 21, 897-904.
3. Wisplinghoff, H.; Bischoff, T.; Tallent, S. M.; Seifert, H.; Wenzel, P.; Edmond, M.B. Nosocomial bloodstream infections in US hospitals: analysis of 24,179 cases from a prospective nationwide surveillance study.Clin. Infect. Dis. 2004, 39, 309-17
The publications listed are very old. It turns out not in the last 20-30 years, but 20-30 years or more ago!!!
New publications should be provided!!!
2. In general, the introduction is too short and relies on publications of data that were also completed a long time ago. There are only 9 links in the “Introduction” section.
The introduction should be significantly expanded by discussing the relevance of the research presented in the work. At the same time, rely on well-known publications over the past five years (2018-2023), bring new publications to the list of references!!! It is possible to discuss the specifics of the problems caused by the COVID-19 epidemic.
3. Clause 2.5. According to the text “The turbidity was confirmed visually and the minimum inhibitory concentration (MIC) determined.”
It should be stated which standard turbidity samples were used.
4. Clause 2.6. According to the text, “the absorbance of the supernatant was measured at 280 nm.”
The device on which these studies were performed should be indicated!
5. Point 4. According to the text “The purpose of this study was not only to identify Candida species and genotyping by DNA sequence analysis, but also to compare antifungal drug susceptibility and protease producibility, which are traits related to pathogenicity. Therefore, wedecided to conduct an investigation using a culture method to obtain viable isolates.”
Question: Why did the authors choose aspartic peptidases for this study?
6. Captions for figures 1-4 are too cumbersome.
Part of the text from the figure caption should be transferred to the text of the article.
7. The article does not have a conclusion.
The text should be supplemented with a conclusion, which sets out the main results of the work and their compliance with the stated goal and objectives.
8. The list of references does not correspond to generally accepted standards. Links to published studies performed in the last 5 years should be provided.
A literature search should be conducted for the last 5 years and the list of publications should be updated.

Author Response

(The authors gave the same response as above.)

Round 2
Reviewer 2 Report
Comments and Suggestions for Authors
All corections improved the quality of the manuscript.
Author Response
Thank you very much for taking the time to review this revised manuscript.